# Medical Text Simplification: Optimizing for Readability with Unlikelihood Training and Reranked Beam Search Decoding

**Lorenzo Flores[1], Heyuan Huang[1], Kejian Shi[1], Sophie Chheang[2], and Arman Cohan[1,3]**

[1]Yale University
[2]Yale School of Medicine
[3]Allen Institute for AI

## Abstract

Text simplification has emerged as an increasingly useful application of AI for bridging the communication gap in specialized fields such as medicine, where the lexicon is often dominated by technical jargon and complex constructs. Despite notable progress, methods in medical simplification sometimes result in the generated text having lower quality and diversity. In this work, we explore ways to further improve the readability of text simplification in the medical domain. We propose (1) a new unlikelihood loss that encourages generation of simpler terms and (2) a reranked beam search decoding method that optimizes for simplicity, which achieve better performance on readability metrics on three datasets. This study's findings offer promising avenues for improving text simplification in the medical field.

## 1 Introduction

In recent years, text simplification has become an increasingly useful application of AI (Stajner, 2021) particularly in healthcare (Carroll et al., 1998; Saggion et al., 2015; Orăsan et al., 2018), where text can be technical and difficult to understand. By automating this process, we can help healthcare professionals explain key medical texts (e.g. doctor's reports, findings) to patients. Previous work in text simplification in medical domain has explored use of pretrained language models (Devaraj et al., 2021; Sun et al., 2023; Martin et al., 2022; Trienes et al., 2022; Basu et al., 2023; Joseph et al., 2023b; Lu et al., 2023), reinforcement learning (Phatak et al., 2022), and zero-shot prompting (August et al., 2022; Joseph et al., 2023b). Despite this progress, simplification sometimes results in the generated text having lower quality and diversity (Devaraj et al., 2021; Phatak et al., 2022). Further as we find some simplification models copy sentences from the source, and thus remain do not sufficiently improve the readability (See Appendix B).

In this work, we seek to further improve medical text simplification. We first propose a new unlikelihood loss that penalizes words in proportion to their reading level using a well-established readability index. Second, we propose a modified beam search method at decoding time to rerank intermediate candidates based on their readability. Despite simplicity, our methods improve readability based on automated metrics (up to 2.43 points on Flesch-Kincaid) and human evaluation, while maintaining similar performance in terms of factual consistency and overall simplification.

We make the following contributions: (1) We propose a new form of unlikelihood loss based on well-established readability index to improve medical text simplification (2) We propose a decoding strategy that optimizes for readability in medical text simplification (3) We provide evaluation results for previous state-of-the-art on three datasets in terms of readability and factual consistency. We make our code publicly available at https://github.com/ljyflores/simplification-project.

**Related Work** Text simplification research primarily focuses on sentence-level (Xu et al., 2015; Specia and Paetzold, 2017; Sulem et al., 2018; Srikanth and Li, 2020; Shardlow and Alva-Manchego, 2022), with some attempts at paragraph or document-level datasets (Sun et al., 2021; Laban et al., 2023). Most datasets have been sourced from accessible Wikipedia or News articles, which are already quite accessible. However, the medical field, laden with technical jargon, can greatly benefit from simplification. Initial methods in medical text simplification employed lexical and syntactic techniques (Llanos et al., 2016; Abrahamsson et al., 2014), while recent work includes finetuning language models like BART (Devaraj et al., 2021; Lewis et al., 2020) and a two-stage summarize-then-simplify approach (Lu et al., 2023). Medical

simplification has also expanded to multilingual settings (Joseph et al., 2023b).

In this work, following Devaraj et al. (2021) we use unlikelihood (UL) training (Welleck et al., 2020) to encourage the generation of simplified terminology. This strategy has been used in other domains to penalize inaccuracy (Hu et al., 2023; Nan et al., 2022), complexity (Devaraj et al., 2021; Lu et al., 2023), and redundancy (Lagutin et al., 2021; Li et al., 2020) in text generation. Unlike Devaraj et al. (2021), our work adapts UL to optimize for both readability and factual consistency. To improve simplification, we also intervene at the decoding stage. Previous work uses modified decoding methods to address factual inconsistency (Shi et al., 2023; King et al., 2022; Sridhar and Visser, 2022), or optimize fluency and diversity in text generation (Kriz et al., 2019; Hargreaves et al., 2021). Our work extends this by optimizing the decoder for readability in medical text simplification.

## 2 Methods

We propose two simple but effective approaches for improving medical text simplification, one during the training phase, and the other during decoding. Specifically, we propose a modified Unlikelihood Loss (Welleck et al., 2020) to incorporate readability index and encourage the model to favor the generation of simpler words. Then, we introduce a decoding approach that evaluates and re-ranks the candidate beams by considering both readability and factuality. We detail these approaches below:

### 2.1 Unlikelihood Loss for Simplification

Unlikelihood loss (UL) (Welleck et al., 2020) is a training objective that forces unlikely generations to be assigned lower probability by the model (See Figure 1).

**Readability UL**  Following prior work (Devaraj et al., 2021) we can use this loss to force the model to assign a lower probability to complex words. Unlike Devaraj et al. (2021), we use the Flesch-Kincaid (FK) readability score (Kincaid et al., 1975) instead of model-predicted scores. The Flesch-Kincaid readability score is a numerical indicator that assesses the complexity of a text by estimating the US grade level needed for comprehension. Because FK considers syllable count and average phrase length, it serves as a good proxy metric even for incomplete sentences, by prioritiz-

ing text with shorter words and shorter phrases. We incorporate this score as follows: At generation step $t$, we identify the word $v$ in the vocabulary with the largest output probability; this is the word which the model is most likely to output at step $t$. We compute the token-level UL for $v$ by taking the product of the word's Flesch-Kincaid score and its standard UL term $log(1 - p(v|\hat{y}_{<t}))$. The total UL ($UL_R$) is the sum of the token-level penalties.

$$UL_R = -\sum_{t=1}^{|\hat{y}|} \sum_{v=1}^{\mathcal{V}} \mathbb{1}_{v,t} FK_v \log(1 - p(v|\hat{y}_{<t}))$$

where $\mathbb{1}_{v,t}$ indicates whether word $v$ has the largest output probability in the vocabulary at step $t$, and $FK_v$ is the Flesch-Kincaid score of word $v$.

**Consistency UL**  As we discuss in §4, we find that $UL_R$ alone leads to hallucinations, hence we also penalize the model for generating unsupported words in some set $e$ using an additional factual consistency UL ($UL_C$).

$$UL_C = -\sum_{t=1}^{|\hat{y}|} \sum_{v=1}^{\mathcal{V}} \mathbb{1}_{v,t} \mathbb{1}_{v,e} \log(1 - p(v|\hat{y}_{<t}))$$

where $\mathbb{1}_{v,e}$ is an indicator for whether the word $v$ is in the set of hallucinated words $e$.

We determine the set $e$ as follows: we identify the sequence which the model is most likely to generate, by finding the tokens with the highest logits at each generation step. Then, we then filter this set to the tokens which do not exist in either the input text nor label. At this point, the set contains all words which the model is likely to generate, but are not present in the input/label. Hence, it may contain words which are factually or grammatically correct, but don't match the gold summary. We'd like to penalize only the tokens which we are sure are factually incorrect, hence we filter this set down to just entities using Spacy en_core_web_lg NER models (Honnibal and Montani, 2017), which results in the entity set $e$.

**Overall Loss**  The overall loss is a weighted sum of the negative log-likelihood ($\mathcal{L}_{NLL}$) and UL, where $\lambda_R$ and $\lambda_C$ are constants.

$$\mathcal{L} = \mathcal{L}_{NLL} + \lambda_R UL_R + \lambda_C UL_C$$

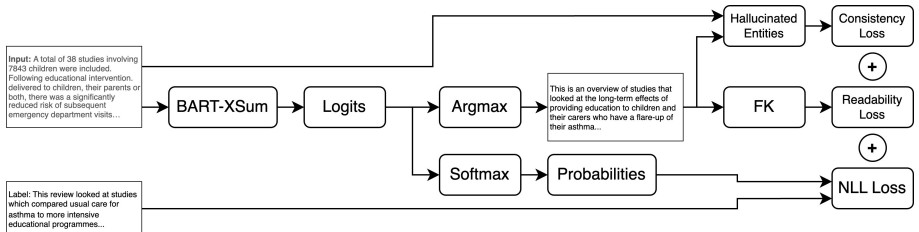

Figure 1: Training Diagram for Computing Unlikelihood Loss

## 2.2 Decoding for Simplification

Our proposed decoding strategy reranks candidate beams by their current readability and factual consistency scores, and retains the top $n$ beams as the candidates for the next token (See Figure 2).

**Readability Score** We optimize candidates' readability during decoding using Flesch-Kincaid (FK) Grade Level scores. FK represents the readability of a text measured by US grade level; hence, lower scores are more readable (Kincaid et al., 1975). These typically range from 0 to 18, but can extend past this range in practice. We compute FK of candidate beams and cap it from 4 to 20, as we find that qualitatively, beams with scores below 4 as equally simple, and above 20 as equally complex. Then, we normalize the score $r_F(s)$ from 0 to 1, such that 0 is least readable, and 1 is most readable.

**Consistency Score** Like in UL training, we find that optimizing solely for readability in decoding may introduce hallucinations; hence we balance readability with consistency, as measured by BERTScore (Zhang et al., 2020). We find that beams with scores below 0.60 to have equally poor factuality, hence we cap the score $r_B(s)$ between 0.60 and 1.00 and normalize it.

**Composite Score** We compute a composite score $r(s)$ using an F1-like metric. Note that the score is merely used to rerank the candidates.

$$r_F(s) = \left\{ \begin{array}{ll} 1, & f_F(s) < 4 \\ \frac{20 - f_F(s)}{20 - 4}, & 4 \leq f_F(s) \leq 20 \\ 0, & f_F(s) > 20 \end{array} \right\}$$

$$r_B(s) = \left\{ \begin{array}{ll} \frac{f_B(s) - 0.60}{0.40}, & f_B(s) \geq 0.60 \\ 0, & f_B(s) < 0.60 \end{array} \right\}$$

$$r(s) = \left( \frac{2 r_F(s) r_B(s)}{r_F(s) + r_B(s)} \right)^2$$

**Ranking Every $k$ Steps** Computing metrics at each generation step can be inefficient, and the meaning or readability of the beam might not change after adding just one word. Hence, we reduce the frequency with which we perform the reranking to intervals of $k$ (See Appendix E).

**Hallucination Heuristic** We implement a heuristic to remove beams with unsupported entities. We identify entities with the Spacy en_core_web_lg NER model (Honnibal and Montani, 2017), check if the entities appear in the source, and set the beam's score as zero if any of the entities are not.

## 3 Experiments

**Datasets** We run our experiments on three datasets: Cochrane (Devaraj et al., 2021) consists of 4,459 pairs of abstracts from the Cochrane Database of Systematic Reviews and their corresponding summaries written by domain experts. MedEasi (Basu et al., 2023) consists of 1,697 pairs of human-annotated sentences sourced from the Merck Manuals (Cao et al., 2020) and SimpWiki (van den Bercken et al., 2019). Finally, the Radiology Reports Dataset [1] consists of 2,269 radiology reports collected from a large urban hospital and simplified by medical residents and doctors.

**Baselines** We compare against a BART-XSum (Lewis et al., 2020) model which we further fine-tune on our datasets, and state-of-the-art models by Lu et al. (2023); Devaraj et al. (2021), all of which we fine-tune on each of the three datasets; we chose BART-XSum to align it with previous work, in order to provide an apples-to-apples comparison and isolate the impact of our methods. We also compare with state-of-the-art large language model GPT-4-0314 (OpenAI, 2023)[2].

**Evaluation Metrics** We evaluate the readability, consistency, and overall performance as follows:

---

[1] Internal dataset

[2] We set the system's role as "You are a helpful assistant that simplifies text", and the prompt as "Simplify this text:".

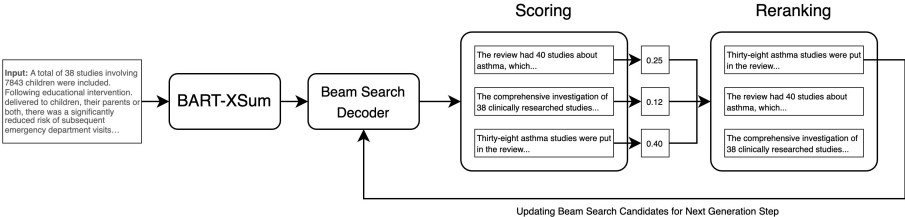

Figure 2: Diagram for Modified Beam Search for Decoding for Simplification

For readability, we use the standard FK (Kincaid et al., 1975) and ARI scores (Smith and Senter, 1967), which use the average word and sentence length to estimate the complexity of texts.

For factual consistency, we use BERTScore (Zhang et al., 2020) and GPT-Eval (Liu et al., 2023) (See Appendix D), as these correlated well with human judgement (Scialom et al., 2021; Li et al., 2022; Liu et al., 2023). For GPT-Eval, we evaluate 50 summaries, and report the fraction of samples in which a factual inconsistency was found.

We additionally use SARI (Xu et al., 2016), an edit-based metric for text simplification, and ROUGE-LSum (Lin, 2004) for overall fluency.

## 4 Results

We fine-tune a BART model using our methods and present the results in Table 1; see Appendix A for implementation details.

**Effect of Unlikelihood Loss and Decoding** On Cochrane and Radiology, our proposed methods achieve better readability scores in terms of FK and ARI. In particular, combining unlikelihood loss with the decoding strategy achieves a 2.43/1.74 point improvement in FK/ARI upon the next best model for Cochrane, and a 0.12/0.17 point improvement for Radiology. Note that in the radiology dataset, the sentences are typically short, resulting in a lower (better) baseline readability score. See sample comparison of outputs in Appendix B.

On MedEasi, our methods slightly underperform NapSS (Lu et al., 2023). We find that it sometimes generates phrases instead of full sentences, which lowers FK/ARI, since these scores depend on sentence length. In contrast, our models generate complete sentences, which improve fluency at the cost of worse (i.e. higher) FK/ARI scores.

Our methods generally improve over the prior SOTA in terms of SARI and BERTScore, however, interestingly on the radiology dataset all methods underperform a fine-tuned BART model.

We observe that using UL or the decoder indi-

vidually results in fewer hallucinations than both BART-UL (Devaraj et al., 2021) and NapSS (Lu et al., 2023) on Radiology, and against NapSS on MedEasi. When the baseline models perform well, we find that it is because they tend to copy information from the input, and hence are less prone to hallucinations. In contrast, our strategies force the model to use simpler words and not copy the input, but may introduce inconsistencies with the source. We confirmed this with an experiment: we compute the % 4-gram overlap of the model written summaries with the source, and observe that large portions of previous works' output is copied from the text, whereas output in our models are not (See Table 3).

Note that some of the identified hallucination errors are relatively minor as we find GPT-Eval to be very strict. For example the phrase "26 self-treatments of 26 Chinese herbal medicine prescriptions" is found to be factually inconsistent with the source having the phrase "26 self concocted Chinese herbal compound prescriptions" by GPT-Eval (see Table 11 for full example).

**Human Evaluation** We conduct a human evaluation study to further investigate the results (See Table 2). We observe that our proposed UL and decoder improves readability over a fine-tuned BART-XSum model 43% and 27% of the time, whereas the previous SOTA NapSS (Lu et al., 2023) only demonstrated clear benefits 3% of the time. However, GPT-4 achieves the best performance, mainly because it is trained on human preference data and omits minor details, only keeping the main summary. In contrast, our models and previous SOTA tend to retain these minor details from the source, which human evaluators may find irrelevant.

We note that the low interrater agreeability aligns with the ranges reported in previous work (Goyal et al., 2023), which reflects the subjective nature of human preference, given that simplicity and readability varies based on one's technical background and style preferences. While such variability is

| Dataset | Model | FK ↓ | ARI ↓ | BScr ↑ | GPT ↓ | SARI ↑ | RL ↑ |
|---|---|---|---|---|---|---|---|
| Cochrane | BART-XSum | 12.19 | 13.83 | 0.871 | **10/50** | 35.64 | **44.76** |
| | GPT-4 | 9.97 | 10.73 | 0.870 | | 39.06 | 33.90 |
| | BART-UL (Devaraj et al., 2021) | 11.02 | 12.69 | **0.873** | 15/50 | 40.08 | 39.25 |
| | NAPSS (Lu et al., 2023) | 12.12 | 13.64 | 0.869 | 21/50 | 32.94 | 45.49 |
| | UL | 8.00 | 9.76 | 0.862 | 27/50 | 42.07 | 40.16 |
| | Decoder | 8.63 | 9.61 | **0.873** | 20/50 | 41.25 | 43.88 |
| | UL + Decoder | **7.54** | **8.99** | 0.866 | 38/50 | **42.12** | 41.11 |
| Radiology | BART-XSum | 3.28 | 2.89 | **0.963** | **19/50** | 78.67 | **80.09** |
| | GPT-4 | 3.85 | 4.41 | 0.862 | | 36.62 | 26.80 |
| | BART-UL (Devaraj et al., 2021) | 2.99 | 2.67 | 0.945 | 28/50 | 69.77 | 68.68 |
| | NAPSS (Lu et al., 2023) | 3.16 | 2.62 | 0.927 | 42/50 | 62.72 | 59.02 |
| | UL | 3.00 | 2.61 | 0.956 | **19/50** | 75.33 | 77.03 |
| | Decoder | 3.11 | 2.76 | 0.952 | 21/50 | 71.75 | 74.57 |
| | UL + Decoder | **2.87** | **2.50** | 0.953 | 23/50 | 72.40 | 74.83 |
| MedEasi | BART-XSum | 10.18 | 11.21 | 0.911 | 28/50 | 40.54 | 45.72 |
| | GPT-4 | 8.10 | 9.20 | 0.903 | | 38.07 | 33.28 |
| | BART-UL (Devaraj et al., 2021) | 10.57 | 11.28 | **0.915** | **2/50** | 35.33 | **47.91** |
| | NAPSS (Lu et al., 2023) | **5.66** | **6.25** | 0.868 | 33/50 | 34.04 | 24.35 |
| | UL | 8.47 | 9.67 | 0.907 | 23/50 | 42.25 | 43.30 |
| | Decoder | 8.27 | 9.66 | 0.908 | 26/50 | **42.66** | 42.91 |
| | UL + Decoder | 7.27 | 9.03 | 0.904 | 31/50 | 41.57 | 40.78 |

Table 1: Performance on Flesch-Kincaid (FK), ARI, BERTScore (BScr), GPT-Eval (GPT), SARI, and ROUGE-LSum (RL); SARI and RL are computed using the EASSE package (Alva-Manchego et al., 2019); All models except for GPT-4 are fine-tuned on the corresponding dataset in the row.

| Model | Readability | $\kappa$ | $\alpha$ |
|---|---|---|---|
| GPT-4 | 93% | 0.190 | 0.199 |
| NAPSS | 3% | -0.118 | -0.105 |
| UL | 43% | 0.004 | 0.0155 |
| Decoder | 27% | 0.236 | 0.245 |

Table 2: Human Evaluation Results on 30 Examples from Cochrane, **Readability** is the % of instances where the model summary was strictly *more* readable than a fine-tuned BART-XSum model's summary, $\kappa$ is Fleiss-Kappa interrater agreement (Fleiss, 1971), $\alpha$ is Krippendorf (Passonneau, 2006).

| Model | % 4-Gram Overlap |
|---|---|
| BART-XSum | 52.88% |
| BART-UL (Devaraj et al., 2021) | 39.30% |
| NAPSS (Lu et al., 2023) | 51.77% |
| UL (Ours) | 15.73% |
| Decoder (Ours) | 9.80% |

Table 3: An analysis of the % 4-gram overlap between the source text and model outputs reveals that previous models tend to copy directly from the source text, whereas our models do not, thereby simplifying and synthesizing

| Model | FK ↓ | BScr ↑ | GPT ↓ | SARI ↑ |
|---|---|---|---|---|
| UL | 8.00 | 0.862 | 27/50 | 42.07 |
| UL ($UL_R$ Only) | 8.74 | 0.863 | 41/50 | 41.37 |
| UL ($UL_C$ Only) | 11.86 | 0.870 | 16/50 | 35.69 |

Table 4: Ablation results on each of the proposed Unlikelihood Losses. Performance on Flesch-Kincaid (FK), BERTScore (BScr), GPT-Eval (GPT), and SARI.

hard to avoid, the average proportions suggest that overall, our methods significantly improved upon previous SOTA (NAPSS).

**Effect of Individual Unlikelihood Losses** We test using $UL_R$ and $UL_C$ separately (See Table 4). $UL_R$ alone results in good readability but poor factual consistency, and vice versa for $UL_C$, justifying the need for both losses to be used in conjunction.

## 5 Conclusion

In this paper, we propose methods to improve simplicity in medical text simplification; this improves the readability of generated summaries, and achieves comparable BERTScore and SARI scores. However, hallucination remains a challenge.

We explored augmenting the data with external knowledge (See Appendix C.2), but found no benefit. This may be because the sources and labels in the training data contains inconsistencies (Lu et al., 2023), which require further preprocessing. Addressing such hallucinations to generate more robust summaries is a critical future direction in medical text summarization, which we aim to explore further.

## Limitations

One limitation of our work is the persistence of hallucinations in the output. Previous literature has shown that this often originates from inconsistencies between the source and text data. For example, a number of training labels in the Cochrane dataset (Devaraj et al., 2021) contain the phrase, "The evidence is up to date as of X", despite no mention of a date in the source (Lu et al., 2023). To this end, future work can adapt strategies from literature in summarization, which have shown that preprocessing (Adams et al., 2022; Wu et al., 2022) and augmenting (Yang et al., 2023) the data can mitigate such hallucinations.

Another limitation is our paper examines medical text simplification very broadly, whereas there may be expert knowledge needed to improve specific tasks. Hence, future work can analyze such methods on a more niche set of datasets (e.g. medical literature, patient reports, health-related news). Such work can be extended to other languages, for which multiple medical text simplification datasets have been developed (Trienes et al., 2022; Grigonyte et al., 2014; Cardon and Grabar, 2019, 2020; Joseph et al., 2023a).

Finally, we note that our inter-annotator agreement on the task of readability is particularly low; this reflects both how human preferences are diverse and how the task is highly subjective, as has been shown in other domains (Goyal et al., 2023). Moreover, readability not only differs by person, but also by domain and task. Future work can define domain-specific criteria, and recruit participants from the exact target populations which the text is meant to be simplified for.

## Ethics Statement

We use publicly available datasets and make our preprocessing and training scripts available. As mentioned in the limitations section, both our methods and previous methods still exhibit varying degrees of hallucination, and have yet to undergo domain-specific examination. Hence, we do not recommend these models be applied in a practical setting at the moment.

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

## A   Implementation Details

We train a baseline BART-XSum model (Lewis et al., 2020) on Cochrane, MedEasi, and the Radiology Dataset. We implement the unlikelihood loss and modified decoder using the Transformers library (Wolf et al., 2020); we report the hyperparameters in Table 5. We run our experiments using NVIDIA-RTX 6000 GPUs.

| Parameter | Value |
|---|---|
| Batch Size | 1 |
| LR | 5e-5 |
| Decay | 0.01 |
| Epochs | 5 |
| $\lambda_R$ | 7.5e-4 |
| $\lambda_C$ | 2.5e-4 |

Table 5:  Training Hyperparameters

## B   Example Output

Tables 6, 7, and 8 show comparisons of outputs from the previous SOTA vs our model. We clearly observe the benefits of our methods; the writing is much simpler, and complex phrases such as "asthma exacerbation" and "emergency department presentation" have been replaced by "asthma attack" and "coming to the emergency department". Table 8 shows an instance wherein the writing is much simpler, but the model tends to retain much more information about the source and explain other concepts (in italics); this may come across as redundant to some evaluators, which explains the results in the human evaluation portion, when compared to models like GPT-4.

## C   Other Experiments

### C.1   Varying Candidate Set $e$

Kindly refer to Table 10 for a comparison of performance when we exclude only words in the input ($UL_I$), the label ($UL_L$), or both ($UL_{IL}$).

### C.2   External Knowledge Augmentation

We identify complex terms using SciSpacy (Neumann et al., 2019), obtain their meanings from UMLS (Bodenreider, 2004), and insert these meanings in the sentence. An example is provided in Table 9. Kindly refer to Table 10 for a comparison of performance with and without external knowledge (EK).

## D   GPT-Eval Implementation

We follow Liu et al. (2023) to implement GPT-Eval for factual consistency with GPT-4, as this was found to correlate best with human judgment. We use the gpt-4-0314 version, with $n$, $top_p$, and $temperature$ set to 1. We set the system role as "Your task is to rate the summary on one metric." For the user input, we input the following:

| Model | Example |
|---|---|
| Input | A total of 38 studies involving 7843 children were included. Following educational intervention delivered to children, their parents or both, there was a significantly reduced risk of subsequent emergency department visits (RR 0.73, 95% CI 0.65 to 0.81, N = 3008) and hospital admissions (RR 0.79, 95% CI 0.69 to 0.92, N = 4019) compared with control. There were also fewer unscheduled doctor visits (RR 0.68, 95% CI 0.57 to 0.81, N = 1009). Very few data were available for other outcomes (FEV1, PEF, rescue medication use, quality of life or symptoms) and there was no statistically significant difference between education and control. Asthma education aimed at children and their carers **who present to the emergency department for acute exacerbations can result in lower risk of future emergency department presentation and hospital admission**. There **remains uncertainty as to the long-term effect of education on other markers of asthma morbidity such as quality of life, symptoms and lung function**. It remains **unclear as to what type, duration and intensity of educational packages are the most effective in reducing acute care utilisation**. |
| NAPSS | A total of 38 studies involving 7843 children were included in this review. Asthma education aimed at children and their carers **who present to the emergency department for acute exacerbations can result in lower risk of future emergency department presentation and hospital admission**. There were also fewer unscheduled doctor visits. Very few data were available for textbfother outcomes (FEV1, PEF, rescue medication use, quality of life or symptoms) and there was no statistically significant difference between education and control. It remains unclear as to what type, duration and intensity of educational packages are the most effective in reducing acute care utilisation. There **remains uncertainty as to the long-term effect of education on other markers of asthma morbidity, symptoms and lung function**. |
| BART-UL | This review of 38 studies involving 7843 children found that asthma education aimed at children and their carers **who present to the emergency department for acute exacerbations can result in lower risk of future emergency department presentation and hospital admission**. There were also fewer unscheduled doctor visits. Very few data were available on other markers of asthma morbidity such as quality of life, symptoms and lung function. It remains **unclear as to what type, duration and intensity of educational packages are the most effective in reducing acute care utilisation**. |
| Ours | This is an overview of studies that looked at the long-term effects of providing education to children and their carers who have a flare-up of their asthma. The aim was to see if there was a benefit in terms of the number of children **who had to go to hospital for an asthma attack**. A total of 38 studies involving 7843 children were included in this review. The results showed that if children were taught about asthma, their parents or both, **they were less likely to come to the emergency department for an attack**. There was also less chance that they would need to go back to hospital. **Very few data were available for other outcomes** (e.g. quality of life, symptoms and lung function) and there was no difference between education and control. However, **there is a need for more research in this area**. |

Table 6: Sample Report 1 from the Cochrane Test Set

| Model | Example |
|---|---|
| Input | Five trials of MSP/RESA vaccine with 217 participants were included; all five reported on safety, and two on efficacy. No severe or systemic adverse effects were reported at doses of 13 to 15 5g of each antigen (39 to 45 5g total). One small efficacy trial with 17 non-immune participants with blood-stage parasites showed no reduction or delay in parasite growth rates after artificial challenge. In the second efficacy trial in 120 children aged five to nine years in Papua New Guinea, episodes of clinical malaria were not reduced, but MSP/RESA **significantly reduced parasite density only in children who had not been pretreated with an antimalarial drug (sulfadoxine-pyrimethamine). Infections with the 3D7 parasite subtype of MSP2 (the variant included in the vaccine) were reduced (RR 0.38, 95% CI 0.26 to 0.57; 719 participants) while those with the other main subtype, FC27, were not (720 participants)**. The MSP/RESA (Combination B) vaccine shows promise as a way to reduce the severity of malaria episodes, but the effect of the vaccine is MSP2 variant-specific. Pretreatment for malaria during a vaccine trial makes the results difficult to interpret, particularly with the relatively small sample sizes of early trials. The results show that blood-stage vaccines may play a role and merit further development. |
| NAPSS | Five trials of MSP/RESA vaccine with 217 participants were included; all five reported on safety, and two on efficacy. No severe or systemic adverse effects were reported at doses of 13 to 15 µg of each antigen (39 to 45 µg total). One small efficacy trial with 17 non-immune participants with blood-stage parasites showed no reduction or delay in parasite growth rates after artificial challenge. In the second efficacy trial in 120 children aged five to nine years in Papua New Guinea, episodes of clinical malaria were not reduced, but the vaccine **significantly reduced parasite density only in children who had not been pretreated with an antimalarial drug (sulfadoxine-pyrimethamine). Infections with the 3D7 parasite subtype (the variant included in the vaccine) were reduced (RR 0.38, 95% CI 0.26 to 0.57; 719 participants)** while those with the other main subtype, FC27, were not (720 participants). The effect of the vaccine is MSP2 variant-specific. Pretreatment for malaria during a vaccine trial makes the results difficult to interpret, particularly with the relatively small sample sizes of early trials. |
| BART-UL | This review of five trials found that the MSP/RESA (Combination B) vaccine **shows promise as a way to reduce the severity of malaria episodes**, but the effect of the vaccine is MSP2 variant-specific. The results show that blood-stage vaccines may play a role and merit further development. |
| Ours | The MSP/RESA (Combination B) vaccine **shows promise as a way to reduce the severity of malaria episodes**, but the effect of the vaccine is MSP2 variant-specific. The results show that blood-stage vaccines may play a role and merit further development. |

Table 7: Sample Report 2 from the Cochrane Test Set

| Model | Example |
|---|---|
| Input | Four studies, involving 125 participants, were included. Three studies evaluated the effects of pulsed electromagnetic fields and one study, capacitive coupled electric fields. Participants with delayed union and non-union of the long bones were included, but most data related to non-union of the tibia. *Although all studies were blinded randomised placebo-controlled trials, each study had limitations. ...* **The overall pooled effect size was small and not statistically significant (risk ratio 1.96; 95% confidence interval 0.86 to 4.48; 4 trials). There was substantial clinical and statistical heterogeneity in this pooled analysis (I2 = 58%). A sensitivity analysis conducted to determine the effect of multiple follow-up time-points on the heterogeneity amongst the studies showed that the effect size remained non-significant at 24 weeks (risk ratio 1.61; 95% confidence interval 0.74 to 3.54; 3 trials), with similar heterogeneity (I2 = 57%). There was no reduction in pain found in two trials. ... More definitive conclusions on treatment effect await further well-conducted randomised controlled trials.**", |
| NAPSS | Four studies, involving 125 participants, were included in this review. Three studies evaluated the effects of pulsed electromagnetic fields and one study, capacitive coupled electric fields. Participants with delayed union of the long bones were included, but most data related to non-union of the tibia. Although all studies were blinded randomised placebo-controlled trials, each study had limitations. The primary measure of the clinical effectiveness of electromagnetic field stimulation was the proportion of participants whose fractures had united at a fixed time point. **The overall pooled effect size was small and not statistically significant (risk ratio 1.96; 95% confidence interval 0.86 to 4.48; 4 trials). There was substantial clinical and statistical heterogeneity in this pooled analysis (I2 = 58%). A sensitivity analysis conducted to determine the effect of multiple follow-up time-points on the heterogeneity amongst the studies showed that the effect size remained non-significant at 24 weeks. There was no reduction in pain found in two trials.**... No study reported functional outcome measures. One trial reported two minor complications resulting from treatment. **More definitive conclusions on treatment effect await further well-conducted randomised controlled trials**. |
| BART-UL | This review of four studies, involving 125 participants, found that electromagnetic field stimulation may offer some benefit in the treatment of delayed union of long bone fractures, but it is inconclusive. **More definitive conclusions on treatment effect await further well-conducted randomised controlled trials.** |
| Ours | We found four studies, with a total of 125 participants. Three of the studies used pulsed electromagnetic fields and one used a type of electric field known as a 'coupled' field. Participants with delayed union and non-union of the long bones were included, but most of the data related to non-unions of the tibia. Although all studies were blinded, *this means that participants did not know whether they were in the treatment group or the control group. Each of the four studies had limitations in the way that it was run and performed.* **The results showed that there was no reduction in pain found in two of the trials. ... Further well-designed, well-conducted randomised controlled trials are required.** |

Table 8: Sample Report 3 from the Cochrane Test Set

| Model | Example |
|---|---|
| Original Text (Cochrane) | A total of 38 studies involving 7843 children were included. Following educational intervention delivered to children, their parents or both... Very few data were available for other outcomes (FEV1, PEF, rescue medication use, quality of life or symptoms)... There remains uncertainty as to the long-term effect of education on other markers of asthma morbidity |
| Text with Context (Context in Red) | A total of 38 studies involving 7843 children were included. Following educational intervention (intended to prevent disease or alter the course of a disease in a patient or population.) delivered to children, their parents or both... Very few data were available for other outcomes (FEV1 (the volume exhaled during the first second of a forced expiratory maneuver started from the level of total lung capacity.), PEF (A synthetic broad-spectrum fluoroquinolone antibacterial agent active against most gram-negative and gram-positive bacteria.), rescue medication use, quality of life or symptoms)... There remains uncertainty as to the long-term effect of education on other markers of asthma (A form of bronchial disorder with three distinct components: airway hyper-responsiveness (RESPIRATORY HYPERSENSITIVITY), airway INFLAMMATION, and intermittent AIRWAY OBSTRUCTION.) morbidity |

Table 9: Labeled Report with Context

| Model | FK ↓ | ARI ↓ | BSc ↑ | S ↑ |
|---|---|---|---|---|
| BART-XS | 11.95 | 12.35 | 0.85 | 36.18 |
| GPT-4 | 9.97 | 10.73 | 0.87 | 39.06 |
| $UL_I$ | 7.74 | 9.47 | 0.86 | 41.93 |
| $UL_L$ | 8.00 | 9.74 | 0.86 | 42.04 |
| $UL_{IL}$ | 8.00 | 9.76 | 0.86 | 42.07 |
| $EK + UL_I$ | 8.82 | 10.82 | 0.85 | 41.18 |
| $EK + UL_L$ | 8.74 | 10.62 | 0.85 | 41.47 |
| $EK + UL_{IL}$ | 8.26 | 9.98 | 0.86 | 41.37 |

Table 10: Performance on Cochrane with External Knowledge (EK), BSc is BERTScore, S is SARI

```
Human Evaluation of Text Summarization
Systems: Factual Consistency: Does the
summary have untruthful or misleading
facts that are not supported by the source
text? Source Text: document Summary:
summary Does the summary contain factual
inconsistencies? Answer:
```

We additionally prompt GPT to explain why it labeled a summary as being factually inconsistent by adding "Why: " after the last line. A sample output is shown in Table 11.

# E   Results by Varying $k$

We reduce the frequency with which we rerank beams, and find that this reduces inference time without large tradeoffs in performance. This was surprising, as we thought reranking less frequently would lead the model to fail to find an "optimal" candidate (See Table 12).

| Model | Example |
|---|---|
| Input | Forty-nine randomised trials involving 3639 participants were included. All trials were conducted and published in China. Thirty-eight different herbal medicines were tested in these trials, including four single herbs (extracts from a single herb), eight traditional Chinese patent medicines, and 26 **self concocted** Chinese herbal compound prescriptions. The trials reported on global symptom improvement (including improvement in numbness or pain) and changes in nerve conduction velocity. The positive results described from the 49 studies of low quality are of questionable significance. There was inadequate reporting on adverse events in the included trials. Eighteen trials found no adverse events. Two trials reported adverse events: adverse events occurred in the control group in one trial, and in the other it was unclear in which group the adverse events occurred. 29 trials did not mention whether they monitored adverse events. Conclusions cannot be drawn from this review about the safety of herbal medicines, due to inadequate reporting. Most of the trials were of very low methodological quality and therefore the interpretation of any positive findings for the efficacy of the included Chinese herbal medicines for treating diabetic peripheral neuropathy should be made with caution. Based on this systematic review, there is no evidence to support the objective effectiveness and safety of Chinese herbal medicines for diabetic peripheral neuropathy. No well-designed, randomised, placebo controlled trial with objective outcome measures has been conducted. |
| Model Summary | Output: Forty-nine randomised trials involving 3639 patients were included in this review. Thirty-eight Chinese herbal medicines were tested in these trials, including four single herbs (extracts from a single herb), eight traditional Chinese patent medicines, and 26 **self-treatments** of 26 Chinese herbal medicine prescriptions. All trials were conducted and published in China. There was no well-designed, randomised, placebo controlled trial with objective outcome measures undertaken. Most of the trials were of very low quality and so the use of any of the included herbal medicines for treatment of neuropathy should be made with caution. |
| Is there inconsistency? Why? | Yes. The summary mentions "26 self-treatments of 26 Chinese herbal medicine prescriptions," which is a slight deviation from the source text which states "26 self concocted Chinese herbal compound prescriptions." Additionally, the summary does not mention the questionable significance of the positive results, adverse events reporting, and the lack of evidence to support the objective effectiveness and safety of Chinese herbal medicines for diabetic peripheral neuropathy. |

Table 11: GPT-Eval rates the summary as being factually inconsistent, even though the summary adequately captures the overall message of the input

| $k$ | $t\downarrow$ | **FK $\downarrow\downarrow$** | **BSc $\uparrow$** | **SR $\uparrow$** | **RL $\downarrow$** |
|-----|------|-------|-------|-------|-------|
| 5   | 20.24 | 9.82  | 0.867 | 40.74 | 42.60 |
| 10  | 19.41 | 9.90  | 0.868 | 40.75 | 42.79 |
| 15  | 19.40 | 10.00 | 0.867 | 40.52 | 42.81 |
| 20  | 18.95 | 9.96  | 0.867 | 40.61 | 42.80 |

Table 12: Performance by $k$ on Cochrane, $t$: Mean Inference Time (seconds), BSc: BERTScore, S: SARI