# OpenReview forum: "Medical Text Simplification: Optimizing for Readability with Unlikelihood Training and Reranked Beam Search Decoding"
_EMNLP/2023/Conference — EMNLP 2023 Findings_

### Official Review · Reviewer_8a31 · 2023-08-03

**Soundness:** 4

**Excitement:**

3: Ambivalent: It has merits (e.g., it reports state-of-the-art results, the idea is nice), but there are key weaknesses (e.g., it describes incremental work), and it can significantly benefit from another round of revision. However, I won't object to accepting it if my co-reviewers champion it.

**Paper Topic And Main Contributions:**

This paper proposes methods to improve the readability of text simplification in the medical domain. The proposed methods show improved performance on readability metrics while maintaining similar performance in terms of factual consistency and overall simplification. However, the issue of hallucinations in the output remains a challenge, and further exploration is suggested to address this issue and improve the robustness of medical text summarization.

**Questions For The Authors:**

Please answer the questions  stated  above.

**Reasons To Accept:**

The paper proposes two simple but effective approaches for improving medical text simplification, one during the training phase and the other during decoding. These approaches incorporate readability index and factuality evaluation to enhance the generation of simplified and accurate medical text.

The human evaluation results demonstrate that the proposed approaches significantly improve readability over a fine-tuned BART-XSum model, outperforming the previous state-of-the-art method. The proposed UL and decoder achieve readability improvements 43% and 27% of the time, respectively, compared to only 3% for the previous SOTA method.

Overall, the paper presents novel approaches for improving medical text simplification, demonstrates their effectiveness through human evaluation, and provides valuable insights for future research directions.

**Reasons To Reject:**

I recommend rejecting this paper for the following reasons:

1. The proposed approaches for improving medical text simplification are relatively simple and do not introduce significant innovations or advancements in the field. The use of unlikelihood training and modified decoding methods have been explored in previous work in other domains. The paper does not provide a strong justification for why these approaches are specifically suitable for medical text simplification or how they address the unique challenges in this domain. Can you provide some insights into this.

2. While the human evaluation results show improvements in readability over a fine-tuned BART-XSum model, the paper does not provide a comprehensive analysis of the limitations and potential biases of the evaluation process. The low inter-annotator agreement on readability and the subjective nature of the task raise concerns about the reliability and generalizability of the evaluation results, which is still discussed in limitations. Without a robust evaluation methodology, it is difficult to determine the true effectiveness of the proposed approaches. If it was previously reported as low, why not devise a different strategy altogether?

**Reproducibility:**

2: Would be hard pressed to reproduce the results. The contribution depends on data that are simply not available outside the author's institution or consortium; not enough details are provided.

**Reviewer Confidence:**

2: Willing to defend my evaluation, but it is fairly likely that I missed some details, didn't understand some central points, or can't be sure about the novelty of the work.

---

> ### Author Rebuttal · Authors · 2023-08-29
>
> Thank you for reviewing our paper and your thoughtful review; we are glad that you found our methods to be simple but effective, and that it hopefully opens up more discussion into such applications in the medical domain.
>
> Thank you for probing deeper into why the methods we propose fit the domain – this is a key point that we will further clarify in the final draft. We also provide additional human evaluation study and share some new findings below:
>
> > 1. The proposed approaches for improving medical text simplification are relatively simple and do not introduce significant innovations or advancements in the field. The use of unlikelihood training and modified decoding methods have been explored in previous work in other domains. The paper does not provide a strong justification for why these approaches are specifically suitable for medical text simplification or how they address the unique challenges in this domain. Can you provide some insights into this.
> * Our approaches address a significant gap in existing methods for medical text simplification: the issue of readability. Prior methods tend to copy from input text and often focus on summarizing and shortening the input. Therefore, while the output is shorter, it can still include technical jargon, resulting in less accessible and readable output. Our techniques, such as UL optimization and reranked decoders, are specifically designed to minimize such direct copying. We added additional analysis that further illustrates this point. In the table below, we observe that previous work has a significantly higher proportion of 4-grams copied from the source. We will include this discussion in the revised manuscript.
>
> | Model | | % of 4-grams Copied from Source |
> | ------- | ----------------- | ---------------------------- |
> | SOTA | BART XSum | 52.88% |
> |           | BART-UL (Devaraj et al., 2021) | 39.30% |
> |           | NAPSS (Lu et al., 2023) | 51.77% |
> | Our Work | UL | 15.73% |
> |                 | Decoder | 9.80% |
>
>
> > 2. While the human evaluation results show improvements in readability over a fine-tuned BART-XSum model, the paper does not provide a comprehensive analysis of the limitations and potential biases of the evaluation process. The low inter-annotator agreement on readability and the subjective nature of the task raise concerns about the reliability and generalizability of the evaluation results, which is still discussed in limitations. Without a robust evaluation methodology, it is difficult to determine the true effectiveness of the proposed approaches. If it was previously reported as low, why not devise a different strategy altogether?
> * We compute Krippendorff’s Alpha to align our method with the work of (Goyal et al., 2023, https://arxiv.org/pdf/2209.12356.pdf) – and find that our range of agreement scores do not vary significantly from theirs (0.05 to 0.18) on a similar task of choosing a preferred summary. This reflects the subjective nature of human preference, as simplicity and readability varies based on one’s technical background and style preferences. While such variability is hard to avoid, the average proportions suggest that overall, our methods (UL and Decoder) significantly improved upon previous SOTA (NAPSS).
>
> | Model | Krippendorff’s Alpha |
> | ------- | ------------------------ |
> | GPT | 0.199 |
> | NAPSS | -0.105 |
> | UL | 0.0155 |
> | Decoder | 0.245 |

---

### Official Review · Reviewer_jGDu · 2023-08-04

**Soundness:** 4

**Excitement:**

4: Strong: This paper deepens the understanding of some phenomenon or lowers the barriers to an existing research direction.

**Paper Topic And Main Contributions:**

They propose a new form of unlikelihood loss based on the readability index to improve medical text simplification.
They also propose a decoding strategy to optimize for readability in text simplification. They evaluated the method on three datasets in terms of readability and factual consistency comparing previous state-of-the-arts.


**Questions For The Authors:**


1. Is it feasible to apply the readability score for each token of the generated sequence at the training and decoding stages?
2. Is there a minimum length of sentence limitation for applying a readability score? If not what kind of tools did you use?
3. In the definition ULc, how the set of hallucinated words e is obtained?

**Reasons To Accept:**

Overall, the paper is well-written and there seem no major issues.
1. The unlikelihood loss based on the readability index is original and intuitive.
2. They compared their method with strong baselines and show the effectiveness of their approach
3. Error analysis with human evaluation adds extra points.


**Reasons To Reject:**

If I don't miss it, there is no information about the available tools to compute Flesch-Kincaid (FK) readability score.
There is no explanation to prove the applicability to use FK score for each token as it seems not intuitive to measure the readability of incomplete sentences.





**Reproducibility:**

4: Could mostly reproduce the results, but there may be some variation because of sample variance or minor variations in their interpretation of the protocol or method.

**Reviewer Confidence:**

3: Pretty sure, but there's a chance I missed something. Although I have a good feel for this area in general, I did not carefully check the paper's details, e.g., the math, experimental design, or novelty.

---

> ### Author Rebuttal · Authors · 2023-08-29
>
> Thank you for taking time to review our paper, we are glad you found our method to be intuitive and effective. We will be sure to clarify the applicability of the Flesch-Kincaid score in the final draft, and provide more details as to its implementation.
>
> > 1. If I don't miss it, there is no information about the available tools to compute Flesch-Kincaid (FK) readability score. There is no explanation to prove the applicability to use FK score for each token as it seems not intuitive to measure the readability of incomplete sentences.
> * Following previous work, we used the EASSE package (Alva-Manchego et al., 2019 – https://aclanthology.org/D19-3009/) to compute FK and provided the implementation in the code repository! The formula takes into account both sentence and word length – thereby making it applicable to both sentences as well as phrases and tokens (i.e. score = 0.39*(total words/ total sentences) + 11.8*(total syllables/ total words) - 15.59). Hence, phrases and tokens with longer words (i.e. more syllables) are deemed more complex. For example, a token with 3 syllables gets a score of 0.39*(1/1) + 11.8*(3/1) - 15.59 = 20.2.
>
> > 2. Is it feasible to apply the readability score for each token of the generated sequence at the training and decoding stages?
> * It is indeed feasible, as FK and ARI both take sequences of any length.
>
> > 3. Is there a minimum length of sentence limitation for applying a readability score? If not what kind of tools did you use?
> * We do not use a minimum sentence length to compute FK, and to the best of our knowledge, there’s no minimum required.
>
> > 4. In the definition ULc, how the set of hallucinated words e is obtained?
> * We define e as the set of entities which are present in the model-generated summary, but not in the source text. Entities are identified using the Spacy NER model (Honnibal and Montani, 2017), applied on the inputs and decoded model summaries. We will clarify this in the revised version.

---

### Official Review · Reviewer_wDdq · 2023-08-04

**Typos Grammar Style And Presentation Improvements:** A figure presenting the method will b…
**Soundness:** 2

**Excitement:**

2: Mediocre: This paper makes marginal contributions (vs non-contemporaneous work), so I would rather not see it in the conference.

**Paper Topic And Main Contributions:**

To simplify medical texts, the paper uses Unlikelihood Loss (UL) and Reranked Beam Search Decoding. The aim is to improve text simplification in the medical domain on readability, factual consistency, and simplicity. The proposed method was evaluated on Cochrane, MedEasi, and an internal radiology reports dataset. The proposed method looks simple (as claimed in line 101) but the results do not support the effectiveness of the method.

**Questions For The Authors:**

1. Why was BART selected?

2. Will the internal dataset be released publicly?

3. Are all three datasets designed for text simplification? or are they created for summarization?

4. How were the annotators selected for human evaluation? Are they non-medical experts?

**Reasons To Accept:**

The paper focuses on text simplification which can be useful in the medical domain, as discussed by the authors, for non-experts to understand expert domain knowledge in layman's languages.

The proposed method is straightforward to understand.

**Reasons To Reject:**

The experimental results cannot support the effectiveness of the proposed methods.

1. The proposed method has consistently poorer performance on the factual consistency (BERTScore and GPT-Eval) in Table 1.

2. The proposed method has a consistently poorer RougeL score in Table 1, which means at the lexicon level, the method is not able to produce simplified text more similar to the gold standard simplified text.

3. The proposed method has poorer readability performance on MedEasi (rather than "better on all three datasets" as claimed in the abstract). As the author also pointed out, the human evaluation of readability has poor inter-annotator agreement.

**Reproducibility:**

2: Would be hard pressed to reproduce the results. The contribution depends on data that are simply not available outside the author's institution or consortium; not enough details are provided.

**Reviewer Confidence:**

4: Quite sure. I tried to check the important points carefully. It's unlikely, though conceivable, that I missed something that should affect my ratings.

---

> ### Author Rebuttal · Authors · 2023-08-29
>
> Thank you for reviewing our paper; we are glad you found our methods straightforward and useful towards laypeople. We appreciate the insightful comments about the quality of our human evaluation and the factual consistency, which prompted us to further scrutinize the outputs qualitatively of previous SOTA.
>
> We provide some of our findings below.
>
>
> > 1. The experimental results cannot support the effectiveness of the proposed methods. The proposed method has consistently poorer performance on the factual consistency (BERTScore and GPT-Eval) in Table 1.
>
> * We note that on multiple datasets, our models achieved the best factual consistency, and had comparable performance on others, showing an overall strong results. Specifically, our decoder variant yielded the highest BERTScore on the Cochrane dataset. On the Radiology and MedEasi datasets, our performance was either comparable or differed by a small margin of at most 0.011 points from competing methods. With respect to GPT-Eval, our UL variant actually tied for best performance on the Radiology dataset with BART-XSum. For the MedEasi dataset, our methods outperform prior work except for BART-UL. While we acknowledge room for improvement on the Cochrane dataset, we would like to highlight the overall strong and competitive performance of our methods across multiple metrics and datasets.
> * We'd like to contextualize our BERTScore and GPT-Eval results by pointing out that while the metrics may suggest a lower score, our analysis reveals that the qualitative nature of the factuality errors is often minor, such as misspellings or paraphrased nouns. These minor discrepancies disproportionately impact automated metrics, as evidenced by existing research (e.g., Guo et al., 2023 – https://arxiv.org/abs/2305.14341). Importantly, our approach offers advantages in simplicity and readability, as detailed in Appendix B.
> * Additionally, we’d like to note that higher factuality scores in previous work are often achieved through extractive methods that copy directly from the source. This inherently boosts factuality but does so at the expense of other desirable traits like readability and originality.
> We added additional analysis that further illustrates this point. In the table below, we observe that previous work has a significantly higher proportion of 4-grams copied from the source.
>
> | Model | | % of 4-grams Copied from Source |
> | ------- | ----------------- | ---------------------------- |
> | SOTA | BART XSum | 52.88% |
> |           | BART-UL (Devaraj et al., 2021) | 39.30% |
> |           | NAPSS (Lu et al., 2023) | 51.77% |
> | Our Work | UL | 15.73% |
> |                 | Decoder | 9.80% |
>
> > 2. The proposed method has a consistently poorer RougeL score in Table 1, which means at the lexicon level, the method is not able to produce simplified text more similar to the gold standard simplified text.
> * While RougeL can gauge overall quality, there are many valid ways of simplifying text which RougeL may not capture as it relies only on lexical overlaps (please see Guo et al., 2023 https://arxiv.org/abs/2305.14341); hence we follow previous work which uses FK and ARI scores as the main metrics for simplicity.
>
> > 3. The proposed method has poorer readability performance on MedEasi (rather than "better on all three datasets" as claimed in the abstract). As the author also pointed out, the human evaluation of readability has poor inter-annotator agreement.
> * We find that NAPSS achieves better FK and ARI scores because it produces shorter phrases which are either incomplete or largely lack context, which artificially lowers FK and ARI (33.66% of outputs had 5 words or less, and median length was 7 words; phrases it generated include “Carcinoma develops”, “Signs should be looked at.”, and “Breaking Professor admits give”). In comparison, our model produces full sentences which improve simplicity (e.g. our UL + Decoder model achieves better FK and ARI than all previous methods except NAPSS) while maintaining fluency. Nonetheless, we will revise the phrase “on all three datasets” in the final draft to more accurately represent the results from the metrics.
> * We calculated Krippendorff's Alpha to assess agreement scores, aligning our methodology with Goyal et al., 2023 (https://arxiv.org/pdf/2209.12356.pdf). The table below shows these results. Our findings reveal that the range of agreement scores do not vary significantly from theirs (0.05 to 0.18) on a similar task of choosing a preferred summary; this could reflect the subjective nature of human preference for this task, as simplicity and readability varies based on one’s technical background and style preferences. We recognize that an in-depth analysis of the complexities and subjectivity of text simplification would be an important contribution, but we believe this is outside the scope of our current work. We will add a discussion about this in the future work.
>
> | Model | Krippendorff’s Alpha |
> | ------- | ------------------------ |
> | GPT | 0.199 |
> | NAPSS | -0.105 |
> | UL | 0.0155 |
> | Decoder | 0.245 |
>
> > 4. Why was BART selected?
> * The previous state-of-the-art methods are implemented on top of BART as the base model. Therefore, to enable direct and fair comparison, we also use BART as the base model. This ensures that we control for other factors and directly evaluate the impact of our proposed unlikelihood loss and decoding approaches.
>
> > 5. Will the internal dataset be released publicly?
> * There are some logistics involved in releasing the data and we are currently looking into this. The dataset is extracted from real world clinical reports in a large academic-cooperated hospital and the simplified reports are written by professional resident doctors.
>
> > 6. Are all three datasets designed for text simplification? or are they created for summarization?
> * Yes, all three datasets are created for text simplification.
>
> > 7. How were the annotators selected for human evaluation? Are they non-medical experts?
> * The annotators selected are non-medical experts, and we deliberately chose non-medical university students, to simulate the target population (i.e. laypeople) of such simplification models
>
> > 8. A figure presenting the method will be helpful.
> * We have drawn a flow chart/ pseudo code table to help better present the logic of our method, which we will include in the appendix.

---

### Meta-Review · Area_Chair_26p1 · 2023-09-23

**Recommendation:** 3

**Metareview:**

The paper proposes two methods to improve simplification of medical texts: unlikelihood loss during training and reranked beam search during decoding. Both techniques intend to  to optimise for readability by relying on flesch-kincaid grade level, which is a metric based on average sentence and word lengths.

Reviewers recognise that the proposed method is straightforward to understand. In addition, the authors compare their approach against strong baselines, and show improvements mostly based on automatic metrics (flesch-kincaid included), as well as with some human evaluation.

One of the main concerns is related to the suitability of flesch-kincaid, traditionally a document-level metric, for assessing the readability of words/phrases and partial sentences. Previous work has pointed out that it is easy to fool this metric (https://aclanthology.org/2021.gem-1.1/) and that it does not agree with judgements of simplicity at the sentence level (https://aclanthology.org/2021.cl-4.28/). However, it is difficult to disregard the results obtained by the authors' experiments, where their proposed model does improve over the BART-XSUM baseline, even using human judgements (despite low inter-annotator agreements, which is common in subjective tasks). In their rebuttal, the authors also point out that their method allows for less conservative outputs, which is an interesting finding that they are recommended to include in the paper.

---

### Decision · Program_Chairs · 2023-10-07

**Decision:**

Accept-Findings

**Comment:**

The paper proposes two methods to improve simplification of medical texts: unlikelihood loss during training and reranked beam search during decoding. Both techniques intend to  to optimise for readability by relying on flesch-kincaid grade level, which is a metric based on average sentence and word lengths.

Reviewers recognise that the proposed method is straightforward to understand. In addition, the authors compare their approach against strong baselines, and show improvements mostly based on automatic metrics (flesch-kincaid included), as well as with some human evaluation.

One of the main concerns is related to the suitability of flesch-kincaid, traditionally a document-level metric, for assessing the readability of words/phrases and partial sentences. Previous work has pointed out that it is easy to fool this metric (https://aclanthology.org/2021.gem-1.1/) and that it does not agree with judgements of simplicity at the sentence level (https://aclanthology.org/2021.cl-4.28/). However, it is difficult to disregard the results obtained by the authors' experiments, where their proposed model does improve over the BART-XSUM baseline, even using human judgements (despite low inter-annotator agreements, which is common in subjective tasks). In their rebuttal, the authors also point out that their method allows for less conservative outputs, which is an interesting finding that they are recommended to include in the paper.